# Mechanical, Structural, and Biological Properties of Chitosan/Hydroxyapatite/Silica Composites for Bone Tissue Engineering

**DOI:** 10.3390/molecules26071976

**Published:** 2021-03-31

**Authors:** Robert Adamski, Dorota Siuta

**Affiliations:** Faculty of Process and Environmental Engineering, Lodz University of Technology, 90-924 Lodz, Poland; robert.adamski@p.lodz.pl

**Keywords:** chitosan, silica, hydroxyapatite, bone regeneration, calcium β-glycerophosphate, sodium β-glycerophosphate pentahydrate

## Abstract

The aim of this work was to fabricate novel bioactive composites based on chitosan and non-organic silica, reinforced with calcium β-glycerophosphate (Ca-GP), sodium β-glycerophosphate pentahydrate (Na-GP), and hydroxyapatite powder (HAp) in a range of concentrations using the sol–gel method. The effect of HAp, Na-GP, and Ca-GP contents on the mechanical properties, i.e., Young’s modulus, compressive strength, and yield strain, of hybrid composites was analyzed. The microstructure of the materials obtained was visualized by SEM. Moreover, the molecular interactions according to FTIR analysis and biocompatibility of composites obtained were examined. The CS/Si/HAp/Ca-GP developed from all composites analyzed was characterized by the well-developed surface of pores of two sizes: large ones of 100 μm and many smaller pores below 10 µm, the behavior of which positively influenced cell proliferation and growth, as well as compressive strength in a range of 0.3 to 10 MPa, Young’s modulus from 5.2 to 100 MPa, and volumetric shrinkage below 60%. This proved to be a promising composite for applications in tissue engineering, e.g., filling small bone defects.

## 1. Introduction

The development of innovative techniques and new biomaterials to fabricate porous, osteogenic, osteoconductive, osteoinductive, non-toxic, and biodegradable implants [1,2] with adequate mechanical strength is a challenge for many scientists, doctors, and engineers in the repair and treatment of bone tissue damaged by cancer, osteomyelitis, congenital defects, or accidents [3,4]. Composites based on ceramics (e.g., bioglasses, silica, hydroxyapatite, and titian), polymers (both natural and synthetic such as chitosan, collagen, fibrin, elastin, alginate, hyaluronic acid, polylactic acid) and hybrid bio-composites [5,6,7,8,9,10,11,12,13] are mainly used in bone tissue engineering applications. Chitosan is an ideal, inexpensive, and readily available copolymer of d-glucosamine and N-acetyl-d-glucosamine [14] derived from the deacetylation of chitin [15] that can be used for the repair and treatment of damaged bone tissue [16]. This polymer is bioadhesive, biodegradable, biocompatible, and cytocompatible [17,18], and supports the immune system by activating macrophages to produce anti-inflammatory cytokines [19,20]. Its low toxic effects and biological inertness have been confirmed in studies, both in vitro and in vivo [21]. However, scaffolds made only with chitosan as the base polymer are characterized by poor mechanical properties including poor tensile strength and low fracture stiffness, fast degradation rate, and low osteoconductivity [4]. Moreover, it is difficult to control pore size during the fabrication of chitosan implants, which play a key role in osteoblast survival, growth, and differentiation [22] as well as transport of essential components necessary for bone regeneration, gas diffusion, and removal of metabolism products. These disadvantages have been overcome by modifying chitosan-based implants with different substances (such as hydroxyapatite, sodium alginate, hyaluronic acid, calcium phosphate, collagen).

Hydroxyapatite (calcium hydroxyphosphate, Ca_10_(PO_4_)_6_(OH)_2_) is a bioceramic material that exhibits chemical and mineralogical similarity to the inorganic component of bones and teeth [23,24]. It is recognized as one of the better implantable materials in bone surgery and dentistry due to its biocompatible, osteoinductive, non-inflammatory, and bioresorbable nature [25]. The structural properties of hydroxyapatite significantly affect the mechanical and biological properties of bone in both in vitro and in vivo studies [26]. However, the degradation rate of synthetic hydroxyapatite is too slow, and the degradation rate of the material substrate may not match the growth rate of the tissue [3]. The addition of hydroxyapatite to the chitosan matrix can improve implant properties such as porosity, water retention, and osteoinductivity [27]. Qiaoling Hu et al. [28] have proved that the higher the hydroxyapatite content in the composite, the more brittle the composite is.

Currently, another promising component that exhibits properties ideal for grafting and scaffolding bone implants is silicon dioxide SiO_2_ or non-organic silica [29,30]. Three organic compounds are very often used as a source of silicate ions: tetraethylorthsilicate (TEOS) [6,31,32], ICPTES (3-isocyanatopropyl triethoxysilane) [31,32], and tetraacetoxysilane (TAS) [5]. Biomaterials based on polymers with silica have higher biomineralization capability than pure polymer. Silica also increases the stiffness of polymer material without significantly causing the mechanical strength of the composite to deteriorate. Silicon plays a significant role in the early stage of cartilage and bone growth and collagen synthesis by osteoblast cells. In addition, the silanol group also promotes the growth of new apatite on the bone substitute surface. Beck et al. [33] proved that silica stimulate bone-forming osteoblasts, suppress bone resorbing osteoclasts, and enhance bone mineral density in vivo. Based on the above properties, it can be expected that silica is an ideal component for improving mechanical properties, cell adhesion, proliferation, and biocompatibility of composite implants for bone regeneration. To our knowledge, biocomposites based on chitosan, hydroxyapatite, inorganic silica, and calcium β-glycerophosphate or sodium β-glycerophosphate pentahydrate have not been studied yet by researchers. Calcium β-glycerophosphate is used for the formation of calcium and phosphorus compounds (mineralization) in implants, which plays a crucial role in the development of both single cells and functional connections between them [34].

The present research is intended to fabricate a porous hybrid bio-composite having optimal bioactive and mechanical strength for bone implants, based on chitosan (CS) and non-organic silica (SI), reinforced with calcium β-glycerophosphate (Ca-GP) or sodium β-glycerophosphate pentahydrate (Na-GP), and hydroxyapatite powder (HAp). The sol–gel method and the convection drying method were applied to the fabrication of implants. The sol–gel method was used because it is highly adaptable and allows control of important properties of implants such as geometry, porosity, and the degree of pores in order to mimic the topological and microstructural characteristics of the extracellular matrix. It is widely known that after drying and rewetting, the initial moisture content of the composite does not return to its original internal structure [35,36]. The reason for this is that during drying the original structure is partially or totally destroyed by the action of surface tension of the solvent removed. Some chemical changes also take place such as the removal of hydroxyl groups from the surface of pores. Nevertheless, drying strengthens the solid, allowing for its storage, and provides the possibility of impregnation with bioactive ingredients. The study also investigated the influence of the drying temperature on the properties of composites. The composites obtained were characterized in terms of chemical and mechanical properties using an Instron universal testing machine, scanning electron microscopy (SEM), and Fourier-transform infrared (FTIR) spectroscopy. The biocompatibility of composites was also assessed.

## 2. Materials and Methods

### 2.1. Materials

Medical grade chitosan powder (CS) from chitin of crab shells with molecular weight of 680 kDa and degree of deacetylation of 80.4% (Sigma-Aldrich, Germany) was used for the preparation of composites. Acetic acid, calcium β-glycerophosphate (Ca-GP), and sodium β-glycerophosphate pentahydrate (Na-GP) were purchased from Sigma-Aldrich (Germany), and hydroxyapatite (HAp, nanopowder, <200 nm particle size) was acquired from Merck KGaA (Germany). Reagents except CS used in experiments were of analytical grade and used as received without further purification.

The source of inorganic silicon was sol of silica (Si, trade name Sizol 030) a commercial product of the “Rudniki” Chemical Works (Poland), essentially composed of 30% SiO_2_ and 0.36% Na_2_O in water, pH ~9.

### 2.2. Methods

#### 2.2.1. Preparation of CS/Si/HAp Composites

This fabrication method of CS/Si/HAp composites intended for bone substitutes is based on the sol–gel technique. Briefly, 0.4 g of chitosan was dissolved in 10 g of 4.0% acetic acid. The solution obtained was stirred (under slow rotations) until complete dissolution for 1 h in a water bath at 55 °C. Next, 3.2–12.7 wt.% of hydroxyapatite was added gradually to the solution and stirred in an ultrasonic bath for 15 minutes to break down powder agglomerates. A total of 5.5 g of CS/HAp mixture was added dropwise to 11 g of silica sol and vigorously stirred. During the synthesis, the pH values of the solutions oscillated between 4 and 8. Then, the paste obtained was poured into cylindrical PE molds with a diameter of 3 cm and height of 3 cm and aged for 24 h. Finally, CS/Si/HAp composites were dried in one day at 50 and 100 °C in the oven at atmospheric pressure. After drying, the cylindrical samples were weighed and measured. Following the fabrication process, the composites obtained were subjected to chemical, mechanical, and biological assessment.

#### 2.2.2. Preparation of CS/Si/HAp/Ca-GP and CS/Si/HAp/Na-GP Composites

The stages for preparation of CS/Si/HAp compositions described in Section 2.2.1. were repeated to obtain different CS/Si/HAp/Ca-GP and CS/Si/HAp/Na-GP composites with various weight ratios of Ca-GP and Na-GP. Briefly, 3.2–12.7 wt.% of hydroxyapatite and 4.0–9.1 wt.% of Na-GP were added gradually to the chitosan–acetic acid solution and stirred in an ultrasonic bath for 15 minutes. Then, CS/HAp/Na-GP mixtures were added to silica sol. The content of SiO_2_ was c.a. 20 wt.%. Next, the pastes obtained were poured into cylindrical PE molds and aged for 24 h. Finally, CS/HAp/Si/Na-GP composites were dried for 24 h at 50 and 100 °C in the oven at atmospheric pressure. In the case of the CS/Si/HAp/Ca-GP composites, the Ca-GP content was 3.3–8.2 wt.%. After drying, the cylindrical samples were weighed, measured, and finally ground to obtain the desired dimensions for the next analysis. To maintain the stability of properties, samples were kept in desiccators with silica gel.

In the studies, the chitosan–acetic acid solution was prepared on the same day as the synthesis of samples. This eliminated the possible impact of degradation of chitosan on the properties of the composites obtained. For a comparison of mechanical properties of the obtained biocomposites, reference samples containing CS/Si, CS/Si/Ca-GP, and CS/Si/Na-GP were also prepared according to proportions set out above, but the content of SiO_2_ was between 22.0 and 26.0 wt.%.

#### 2.2.3. Determination of Density and Volumetric Shrinkage

Densities of dry samples (*ρ*_s_) of CS/Si, CS/Si/HAp, CS/Si/HAp/Ca-GP, CS/Si/Ca-GP, CS/Si/HAp/Na-GP, CS/Si/Na-GP were determined based on mass (m_s_) and volume of dry gels (V_s_), according to Equation (1):(1)$$\rho_{s}=\frac{m_{s}}{V_{s}}$$

Skeletal densities were also determined using a helium pycnometer (AccuPyc 1330, Micromeritics Instrument Corporation, Norcross, GA, USA).

The volumetric shrinkage coefficient (σ) as a percentage (%) for each sample was calculated using the formula (Equation (2)),
σ = (1 − V_s_/V_m_)·100%(2)
where V_s_ is the volume of the dry cylindrical sample and V_m_ is the volume of the wet sample.

#### 2.2.4. Mechanical Testing of Composites

Mechanical properties of CS/Si, CS/Si/HAp, CS/Si/HAp/Ca-GP, CS/Si/Ca-GP, CS/Si/HAp/Na-GP, and CS/Si/Na-GP biocomposites were investigated at room temperature using a computer-controlled universal testing machine (Instron 2519-107, USA) at maximum load 5 kN and constant speed of crosshead displacement 0.2 mm/min. Before the tests, each cylindrical sample was ground after drying to align the upper and lower surface. The initial length of the specimens was 15 ± 0.1 mm. A mean value of at least five different measurements was obtained, and a standard deviation was calculated. Young’s modulus values were determined from the slope of the linear part of stress–strain curves, and compressive strength was obtained from the first maximum of stress visible in the curves.

#### 2.2.5. Surface Analysis of Composites

The surface morphology of CS/Si, CS/Si/HAp, CS/Si/HAp/Ca-GP, CS/Si/Ca-GP, CS/Si/HAp/Na-GP, and CS/Si/Na-GP nanocomposites was investigated using scanning electron microscopy (FEI model Quanta 200F, Thermo Fisher Scientific, Hillsboro, OR, USA), coupled with a field emission gun (FEG) and energy dispersive spectroscopy (EDS) equipment (EDAX model Genesis 4000). Experiments were performed under a nitrogen atmosphere at a pressure of 100 Pa (low vacuum operating mode). This mode avoids coating the sample with a thin conductive layer, such as gold or carbon, which is important in order not to distort the surface topography.

#### 2.2.6. Structure Characterization of Composites

Attenuated total reflection Fourier-transform infrared (ATR–FTIR) spectroscopy was applied to characterize intermolecular interactions between components included in the composites. The spectra of the pure components used in synthesis and spectra for the prepared composites were recorded. The ATR–FTIR spectra were recorded in the 4000 to 650 cm^−1^ range using a Jasco FT/IR 6200 spectrometer (JASCO Inter. Co., Ltd., Tokyo, Japan) equipped with an MCT M detector cooled by liquid nitrogen (77K) and a MIRacle ATR sampling accessory (diamond/ZnSe) (PIKE Technol., Fitchburg, WI, USA). The whole spectrometric system was purged by dry argon. The interferometer scanning rate was 0.1 cm/s. Signal accumulation from 300 scans was taken with a resolution of 1 cm^− 1^. The test specimens were in the form of powder, without previous compression.

#### 2.2.7. Biocompatibility Analysis of Compositions

KYOU DXR0109B human induced pluripotent stem (IPS) cells [201B7] (ATCC^®^ ACS1023™; Manassas, VA, USA) derived from fibroblasts obtained from a healthy donor were used for the study. Prior to testing, all culture vessels and samples of the tested composites were coated with CellMatrix™ Basement Membrane Gel, (ATCC ACS3035) to enable cell adhesion, according to the guidelines provided by the vendors.

The culture was carried out in Eppendorf tubes (cell culture plates, size 12 wells, surface treatment TC treated, flat bottom clear wells) on 12-well plates. An implant was placed from each type of test composite used in standard culture vessels, and then a 20,000 cells/cm^2^ area was flooded with 2 mL of the Pluripotent Stem Cell SFM XF/FF culture medium (ATCC ACS3002). The culture was carried out in an incubator (PHCBI CytoGrow incubator, Panasonic) at 37 °C, 95% humidity, in the presence of 5% CO_2_. Cells were incubated for 7 days (according to the ATCC^®^ recommendation, KYOU DXR0109B cell culture time between passages) in the presence of the test biomaterials relative to a control culture under standard conditions. The culture medium was changed daily, and the suspended cells present in the medium were harvested, counted, and evaluated for viability. Cells were harvested using a stem cell dissociation reagent (ATCC ACS3010), and the number of viable cells was assessed using a Trypan Blue assay (BioRad) and a TC20 ™ Automated Cell Counter (BioRad). Statistical analysis was performed using Student’s t-tests for independent samples and the ANOVA method. Tests were repeated three times for each sample and control.

## 3. Results and Discussion

### 3.1. Morphological Characteristics

The sol–gel method applied allows bone composites to be obtained. Examples of CS/Si/HAp, CS/Si/HAp/Na-GP, CS/Si/Na-GP, and CS/Si/HAp/Ca-GP samples can be seen in Figure 1.

The resulting composites had a three-dimensional structure, but the color, porosity, and stiffness were different. The color of the implants varied; the hydroxyapatite-based implants were whiter than the non-hydroxyapatite implants. A higher concentration of hydroxyapatite resulted in a greater porosity of the structure and thus a more complex external topography. Appropriate surface chemistry and a highly porous structure of the implants are needed to support cell growth, adhesion, proliferation, oxygen, and nutrient diffusion and remove metabolic waste produced during the bone tissue regeneration process. Porosity more significant than 75% is required to obtain osteoconductive properties of composites [3]. In order to assess the structural differences of the implants, photos were taken using SEM. Figure 2 presents example SEM images of the structures obtained of CS/Si/HAp/Ca-GP, CS/Si/HAp/Na-GP, CS/Si/HAp, and CS/Si composites that can be applied for bone tissue regeneration. The SEM image control showed that the surface of CS/Si (Figure 2d) was smooth, while those of CS/Si/HAp/Na-GP (Figure 2b) and CS/Si/HAp (Figure 2c) were rough with slots with pores size below 10 µm. However, the surface of the CS/Si/HAp/Ca-GP (Figure 2a) composite was rough, with corrugation and cornification with a well-developed microstructure. This composite is characterized as having pores of two sizes: large ones of 100 μm and many smaller pores below 10 µm, the behavior of which should positively influence cell proliferation and growth.

### 3.2. Mechanical Assessment

The mechanical properties of the implants after drying at 50 and 100 °C were investigated at ambient temperature. The effect of the HAp, Na-GP, and Ca-GP contents on the mechanical properties of hybrid composites was analyzed. Young’s modulus, strain, and compressive strength of CS/Si/HAp composites are presented in Figure 3. The mechanical properties of CS/Si/HAp composites in terms of Young’s modulus significantly increased when increasing the concentration of HAp in the sample and reached a maximum concentration of 1.0 wt.%. The value of modulus decreased, suggesting a poor interfacial bonding between HAp particles and CS/Si molecules. The results of compression tests showed that the addition of HAp to CS/Si composite promoted compressive strength while causing a decrease in elongation at break. The highest value of Young’s modulus of 25 MPa for CS/Si/HAp composites was obtained for drying samples at 100 °C and 71 MPa for 50 °C.

In the study, the influence of drying temperature on the mechanical properties of the developed composites was noted and is shown in Figure 3. The results obtained of Young’s modulus in case of drying the sample at a temperature of 100 °C were lower compared to 50 °C. In the bar graph in Figure 4, the Young’s modulus values for CS/Si/Na-GP and CS/Si/HAp/Na-GP composites are compared. Analyzing results for the samples of CS/Si/Na-GP (1–4), we can see that the higher concentration of Na-GP in a sample, the lower the value of Young’s modulus.

A similar situation was observed for CS/Si/HAp/Na-GP composites, but this case is more complicated. During the fabrication of CS/Si/HAp/Na-GP and CS/Si/Na-GP composites, the concentrations of SiO_2_ and chitosan were at the same level, but concentrations of HAp and Na-GP were different. Though the CS/Si/HAp/Na-GP composites exhibited better mechanical properties than CS/Si/HAp or CS/Si/Na-GP, the presence of Na-GP weakened the composite. Based on the results presented in Figure 4, it can be concluded that in samples with higher concentrations of HAp, Young’s modulus was higher than in samples with higher concentrations of Na-GP. The peak value of Young’s modulus for these composites was 66.44 ± 3.32 MPa.

A completely different trend was observed for the composites doped with Ca-GP. The values of Young’s modulus increased after mixing with Ca-GP, but this trend heavily depended on the concentration of both Ca-GP and HAp as illustrated in Figure 5. For samples with Ca-GP without HAp, a twofold increase in the concentration of Ca-GP causes a slight increase in Young’s modulus. On the other hand, for composites with 7.7 wt.% of Ca-GP and 5.0 wt.% of HAp, the peak value of Young’s modulus of 100.75 ± 4.99 MPa was determined in the group of all tested configurations.

For comparison, the work of Sowjanya [37] presented composites based on chitosan, reinforced with nanosilica powder and sodium alginate, which were characterized by compressive strength in a range of 0.59 to 0.66 MPa and Young’s modulus of 8.16 to 8.99 MPa. Composites obtained in the present work were characterized by compressive strength in a range of 0.3 to 15 MPa, while the cancellous bone is characterized by a modulus of 0.8 to 1000 MPa, and the compressive strength of human bone is 1 to 210 MPa. The Young’s modulus results obtained for CS/Si/HAp, CS/Si/HAp/Ca-GP, CS/Si/Ca-GP, CS/Si/HAp/Na-GP, CS/Si/Na-GP were compared to CS/Si reference composites. The composition and properties of CS/Si are presented in Table 1. These samples were characterized by a very low Young’s modulus and compressive strength.

The content of HAp and/or x-GP (where x is Na or Ca) to chitosan has a strong effect on the mechanical properties of implants. All the above results suggest that alteration of mechanical properties is a consequence of the interaction between chitosan, silica sol, and components based on calcium. Better mechanical properties can be a result of crosslinking reactions within chains of chitosan and molecular interaction between biopolymer and silica, which stiffens the skeleton of the composite.

### 3.3. Density and Volumetric Shrinkage

Densities of dry samples of CS/Si, CS/Si/HAp, CS/Si/HAp/Ca-GP, CS/Si/Ca-GP, CS/Si/HAp/Na-GP, and CS/Si/Na-GP were calculated using Equation (1), and volumetric shrinkages were estimated using Equation 2. Densities obtained of CS/Si, CS/Si/HAp, CS/Si/HAp/Ca-GP, CS/Si/Ca-GP, CS/Si/HAp/Na-GP, and CS/Si/Na-GP implants were in the range of 320 to 820 kg/m^3^, which is comparable to density of cancellous bone which ranges from 300 to 2100 kg/m^3^ depending on age of the person. The drying temperature had an impact on the results of the density and volumetric shrinkage of the materials obtained. Drying samples at 100 °C reduces the volume shrinkage by 50% as shown in Figure 6 and increases the density of the samples compared to drying at 50 °C. In addition, the volumetric shrinkage of samples also depends on content and composition and is within a range of 5% to 45%. The experimental data obtained are difficult to describe by explicit functional dependence. An additional series of measurements to characterize the trend in the course of the volumetric shrinkage needs to be performed. In most cases, composites with Na-GP characterized the lowest volumetric shrinkage compared with composites of CS/Si, CS/Si/HAp, CS/Si/HAp/Ca-GP, and CS/Si/Ca-GP.

CS/Si/HAp, CS/Si/Ca-GP, CS/Si/HAp/Ca-GP implants were chosen as the most promising composites for further FTIR analysis and biological studies.

### 3.4. FTIR Analysis

The molecular interactions of CS/Si/HAp, CS/Si/Ca-GP, and CS/Si/HAp/Ca-GP implants were studied by FTIR analysis. Figure 7 shows spectra of composites obtained and represents the range of individual components used in fabrication. Differences are observed between pure components and composites.

The spectrum obtained for chitosan powder samples shows typical bands, such as a broad band in the region of 3500–3000 cm^−1^ corresponding to an overlap of OH group vibration and NH group vibration stretching. A characteristic peak at 2919.2 cm^−1^ is indicative of the C-H bond vibration antisymmetric stretching. An N-H chemical bond in the primary amine and secondary amide occurred at 1540 cm^−1^. Deforming, antisymmetric vibrations for the C-H bond were confirmed by a peak at 1458 cm^−1^ and stretching vibration for N-C-O by a peak at 1241.5 cm^−1^. The band for the C-O group vibration stretching (amide I band) was confirmed by a peak at 1651.2 cm^−1^, and vibration stretching for the C-O-C group by a peak at 1063.07 cm^−1^.

The infrared spectra of raw HAp powder exhibit characteristic absorption bands such as a broad band in the region of 3500–3000 cm^−1^ with a peak at 3245 cm^−1^ corresponding to the vibration of the OH group as a result of water in hydroxyapatite and adsorbed by components from the atmosphere. P-O stretching vibration at 1236 cm^−1^ and P-O asymmetric stretching vibration (the PO_4_^3−^ ν1 mode) appeared at 1026 cm^−1^.

For spectra of Sizol, a peak at 3270 cm^−1^ for the surface OH group can be observed. The presence of the Si-H stretching bond is confirmed by the peak at 2154 cm^−1^ and that at 1062.59 cm^−1^, equivalent to Si-O-C stretching bond (open chain) or Si-O-Si symmetric stretching bond vibration.

The Ca-GP spectrum showed that for a peak at 1250 cm^−1^, C-C skeletal vibrations overlap with the P-O stretching bond. The C-O and P-O-R stretching bands were confirmed by peaks at 1253 and 1005 cm^−1^.

The composite CS/Si/Ca-GP shows several differences compared with the spectra of pure components. The band at 2020.55 cm^−1^ is less pronounced, while bands in the region of 1770–1320 cm^−1^ become sharper. In spectra for a composite, the peaks at 1241.45 and at 946.4 cm^−1^ are less distinct. The band at 2415.89 cm^−1^ (corresponding to O-H group stretching vibration) disappears completely, while the band at 2558.59 cm^−1^ becomes more pronounced (N-H in amine salts). This could indicate that Si atoms join the group OH in chitosan. The peak at 1061.14 cm^−1^ is more significant and smooth. Furthermore, the band shift from 1063.07 to 1061.14 cm^−1^ indicates some interaction in symmetric bond vibration Si-O-Si.

For composites CS/Si/HAp, FTIR spectra show similar differences as described above for CS/Si/Ca-GP composites. A new peak appears at 2557.15 cm^−1^, the peak at 2415.88 cm^−1^ disappears, and the peak at 1049.57 cm^−1^ is more significant. Furthermore, a double peak appears instead of a single peak at 669.41 cm^−1^. The band shift from 1063.07 to 1049.57 cm^−1^ indicates some interaction with the symmetric bond Si-O-Si. The band shift is also observed for OH group vibration, and the peak for this group in the CS/Si/HAp composite is at 3261.52 cm^−1^ (for CS at 3245.61 cm^−1^, Si at 3270.68 cm^−1^, and HAp at 3244.65 cm^−1^).

The CS/Si/HAp/Ca-GP composite exhibited similar behavior. The band at 2560.56 cm^−1^ becomes more pronounced. A significant band shift is also observed for OH group vibration, and the peak is at 3260.56 cm^−1^ (where for Ca-GP it is at 3240.31 cm^−1^ and the peak is flatter than for other pure components and composites). In the case of Si-O-Si, a shift is observed from 1063.07 to 1035.10cm^−1^.

Comparing spectra of composites, for CS/Si/Ca-GP and CS/Si/HAp we can observe a double peak at 670cm^−1^, while for CS/Si/HAp/Ca-GP for the same wave number a single peak is observed. The bands in the region of 1800–1300 cm^−1^ are sharper for CS/Si composites than for other composites analyzed. In each case, the band shift for OH band vibration and Si-O-Si vibration indicates interaction between the Si atom and the OH group. The highest value for this shift is obtained when Ca-GP is added to the composite, and for this sample, the calculated value of Young’s modulus was the highest. The new bond between Si and the OH group reinforces the composite.

In all probability, a very high concentration of Sizol in composites, in comparison with other components, and the possibility of precipitating crystals can impair molecular interactions.

### 3.5. In Vitro Study

The composite’s essential characteristics, determining its suitability for bone tissue applications, are its biocompatibility and cytotoxicity, which require both in vitro studies on isolated, well-defined cells or tissues in laboratory conditions and in vivo testing on animals. In vitro tests are inexpensive and allow initial results of the cell reaction to the tested composite to be obtained quickly. In this study, fibroblast cells were used in cytotoxicity and biocompatibility tests of CS/Si/HAp, CS/Si/Ca-GP, and CS/Si/HAp/Ca-GP composites. Results obtained of cell growth, proliferation, and viability are displayed in Figure 8.

The growth of fibroblasts after 7 days of cell culture was compared to cells cultured without composites (control, *p* > 0.05). The lowest mean value of cell growth (Figure 8a) among the composites tested was recorded in a well with CS/Si/HAp (149%), but the differences are not statistically significant compared to the control. The highest cell growth was observed in the wells with CS/Si/HAp/Ca-GP composite due to its good mechanical properties, highly porous microstructure (greater than 75%), and surface roughness. There were additional perforations in the walls of the pores CS/Si/HAp/Ca-GP composite, which are necessary for proper cell growth, cell migration, and mass transport. The cell proliferation assay indicated that none of the CS/Si/HAp, CS/Si/Ca-GP, or CS/Si/HAp/Ca-GP composites caused a cytotoxic effect, confirmed by the lack of cell death in comparison to the control (100% of viability). The highest cell viability and proliferation were also observed in the CS/Si/HAp/Ca-GP composite. The results obtained of structural, mechanical, and biological investigation of composites showed that the CS/Si/HAp/Ca-GP composite has promising potential in biomedical applications such as bone regeneration, and therefore animal tests can be considered.

## 4. Conclusions

In this study, a new, patented method of fabrication of bio-hybrid Cs/Si, CS/Si/HAp, CS/Si/HAp/Ca-GP, CS/Si/Ca-GP, CS/Si/HAp/Na-GP, and CS/Si/Na-GP composites using the sol–gel technique was developed. The basic structural, mechanical, and biological properties of composites obtained were investigated through FTIR, SEM, and the Instron universal testing machine. Of all composites analyzed, the CS/Si/HAp/Ca-GP composite obtained was the most promising for further research in the field of biomaterials for bone tissue regeneration. It was characterized by the well-developed surface of pores of two sizes: large ones of 100 μm and many smaller pores below 10 µm, the behavior of which positively influenced cell proliferation and growth, as well as a density comparable to the bone density, compressive strength of 0.3–10 MPa, Young’s modulus of 5–100 MPa, and volumetric shrinkage below 60%. The CS/Si/HAp/Ca-GP composite had an appropriate morphology and did not show cytotoxicity towards the tested fibroblasts. Positive test results predispose the material to be suitable for animal testing.

The study showed significant dependence on the drying temperature mainly on the resulting structure’s physicochemical properties of composites. Drying samples at 100 °C reduced the volume shrinkage by 50% and increased the density and mechanical properties of the samples compared to drying at 50 °C. However, it was noticed that the use of high-temperature convection drying at atmospheric pressure is an advantage because it reduces the costs of obtaining the biomaterial and can also replace the sterilization process. In future research, drying strategies should be designed to achieve optimal physicochemical properties of the implants for bone tissue regeneration.

## Figures and Tables

**Figure 1 molecules-26-01976-f001:**
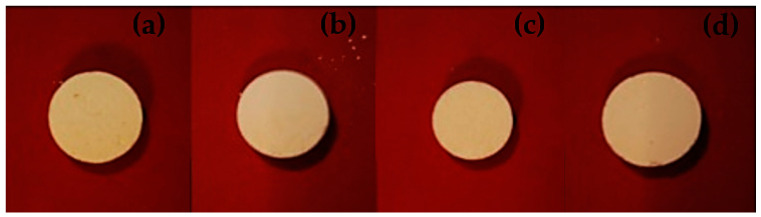
Example photograph of composites fabricated by the sol–gel method (**a**) CS/Si/HAp; (**b**) CS/Si/HAp/Na-GP; (**c**) CS/Si/Na-GP; (**d**) CS/Si/HAp/Ca-GP.

**Figure 2 molecules-26-01976-f002:**
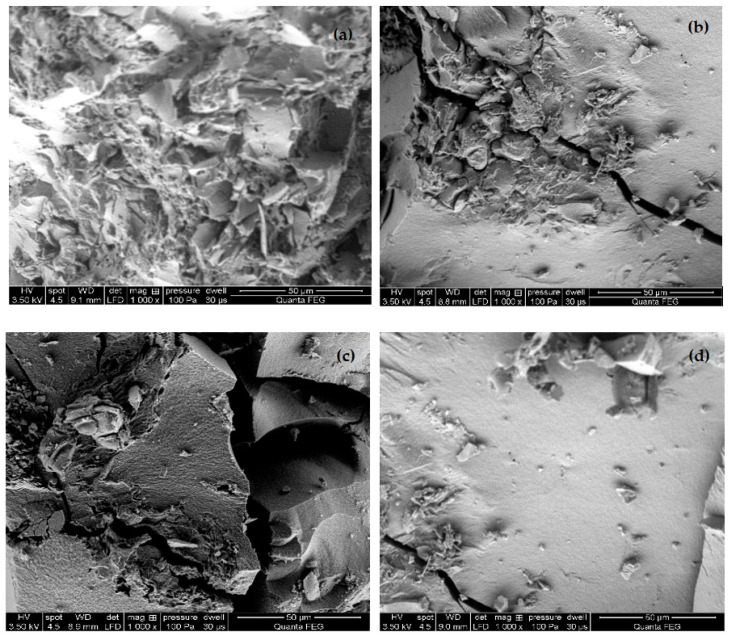
Example SEM images of (**a**) CS/Si/HAp/Ca-GP; (**b**) CS/Si/HAp/Na-GP; (**c**) CS/Si/HAp; (**d**) CS/Si composites with a similar concentration of HAp.

**Figure 3 molecules-26-01976-f003:**
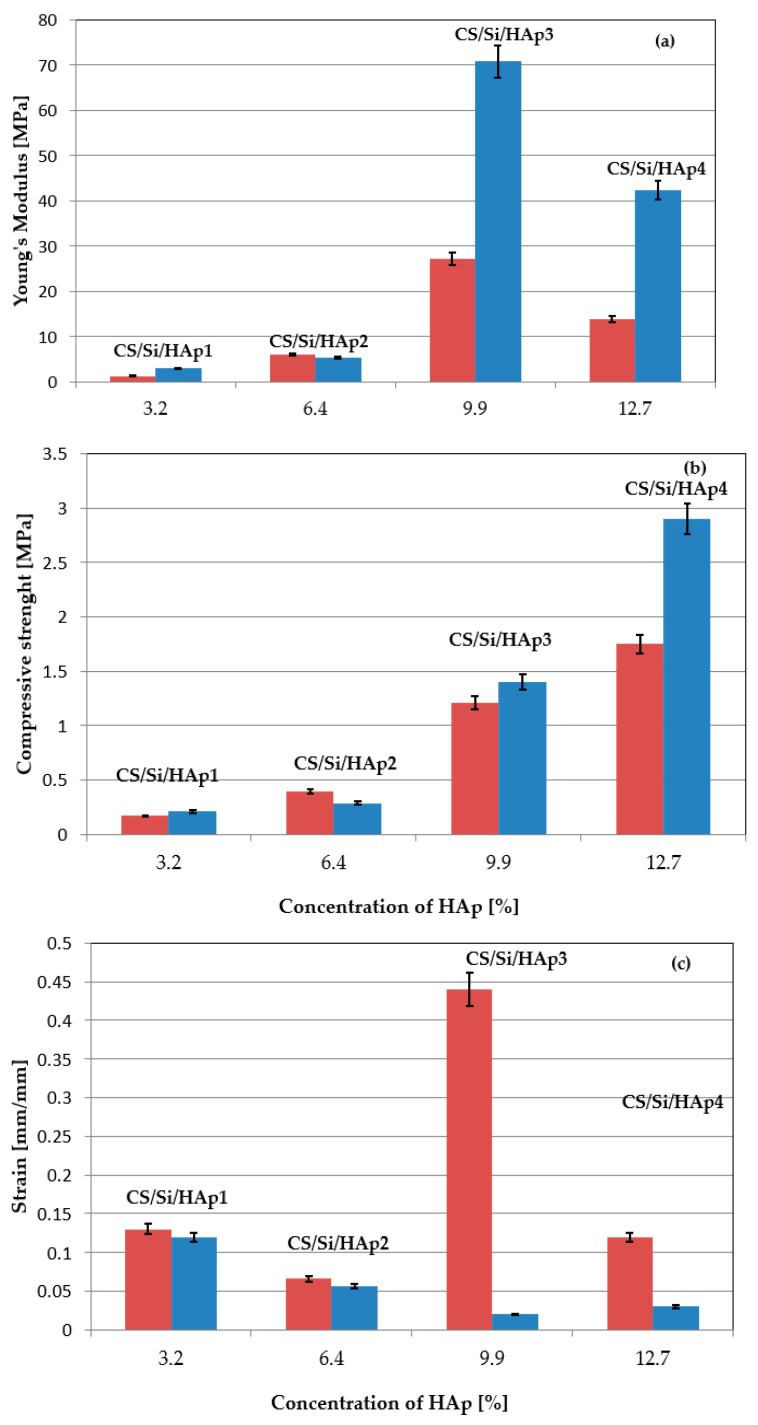
Dependence of Young’s modulus (**a**), compressive strength (**b**), strain (**c**) vs. concentration of hydroxyapatite with 20 wt.% SiO_2_ and 0.7–1.1 wt.% chitosan. The drying of samples at a temperature of 100 °C is marked in blue and at 50 °C in red.

**Figure 4 molecules-26-01976-f004:**
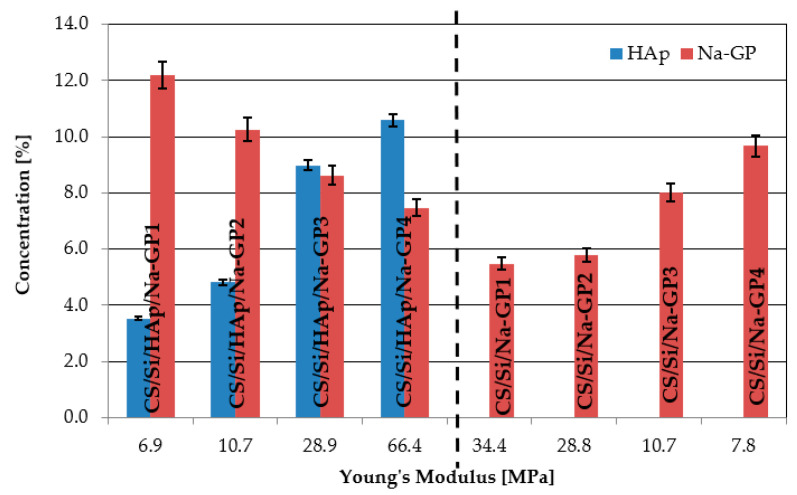
Influence of HAp and Na-GP concentrations on Young’s modulus (labels above each composite) for drying samples at 100 °C.

**Figure 5 molecules-26-01976-f005:**
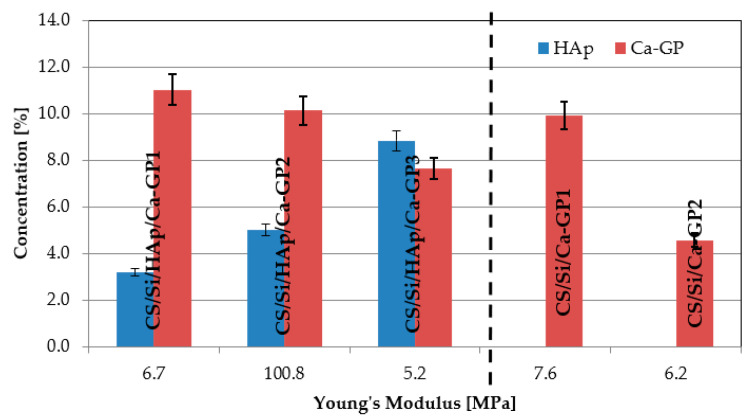
Influence of hydroxyapatite and Ca-GP on Young’s modulus (labels above each composite) for drying samples at 100 °C.

**Figure 6 molecules-26-01976-f006:**
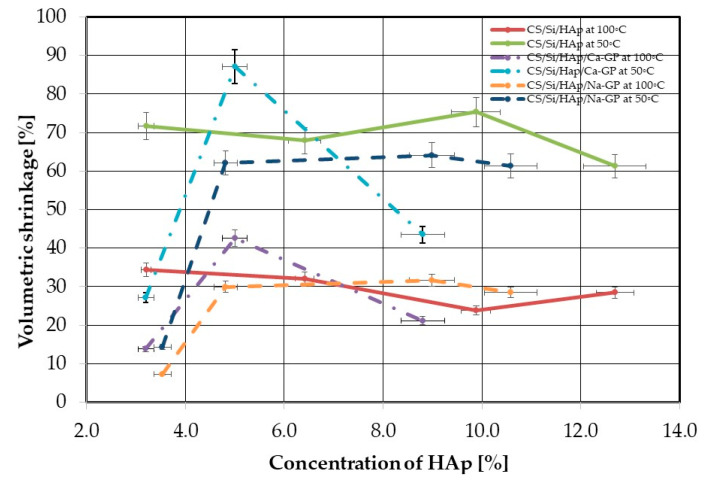
Volumetric shrinkage of CS/Si/HAp, CS/Si/Ca-GP, CS/Si/HAp/Ca-GP composites at drying temperature of 50 and 100 °C.

**Figure 7 molecules-26-01976-f007:**
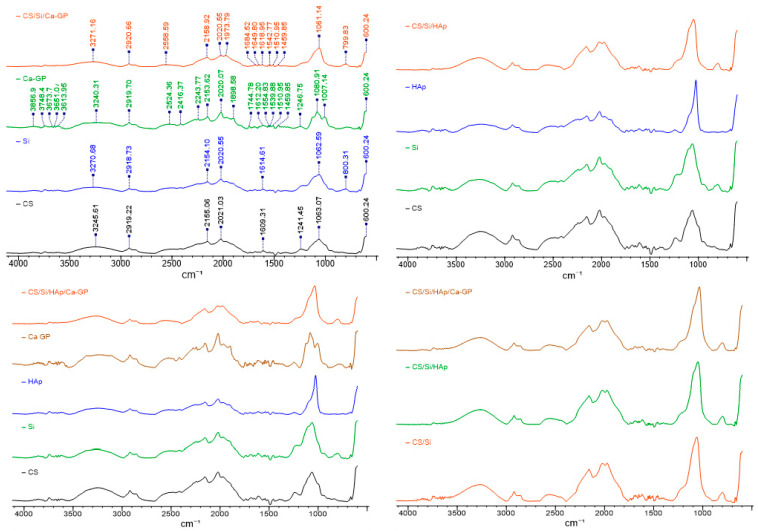
FTIR characterization of implants made of CS/Si/HAp, CS/Si/Ca-GP, CS/Si/HAp/Ca-GP.

**Figure 8 molecules-26-01976-f008:**
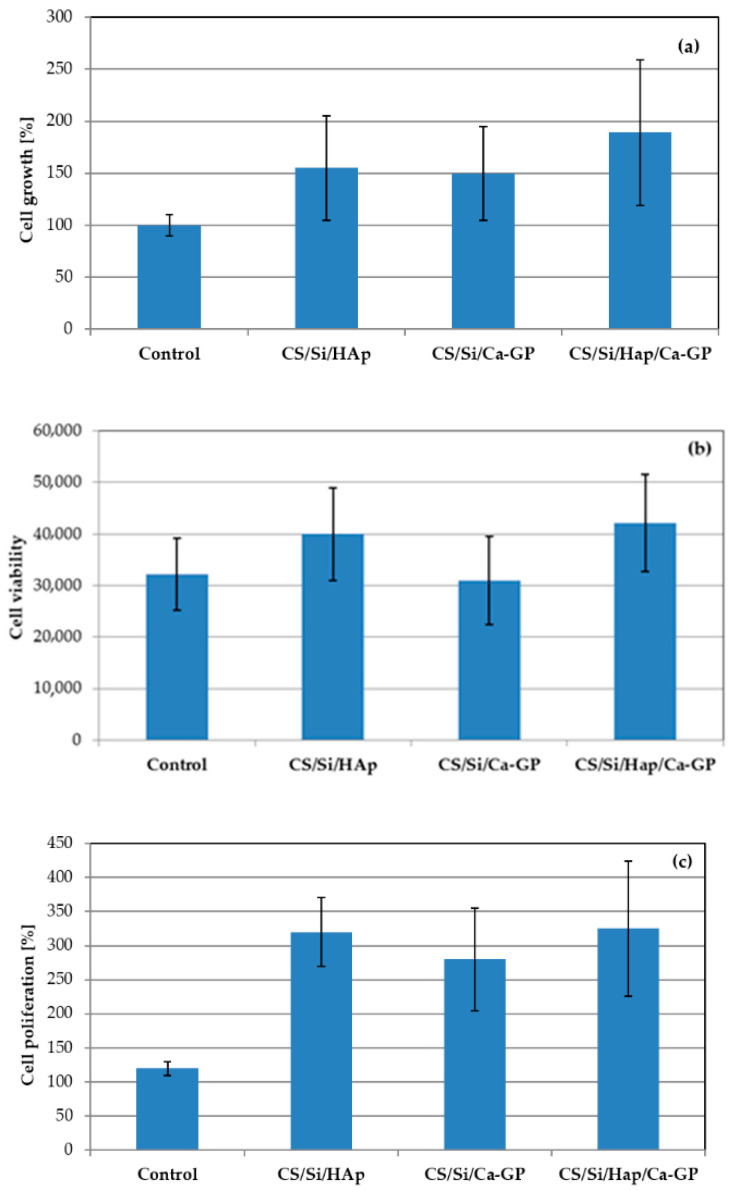
Cell growth (**a**), viability (**b**), and proliferation results (**c**) of CS/Si/HAp, CS/Si/Ca-GP, CS/Si/HAp/Ca-GP composites.

**Table 1 molecules-26-01976-t001:** Properties of references samples of CS/Si.

Sample	Chitosan (wt.%)	SiO_2_ (wt.%)	Young’s Modulus (MPa)	Compressive Strength (MPa)	Strain (mm/mm)	r_s_ (g/cm^3^)	V_s_/V_m_ (%)
CS/Si 1	0.66	24.84	0.39 ± 0.04	0.016 ± 0.004	0.04 ± 0.001	0.40 ± 0.04	3.5 ± 0.32
CS/Si 2	0.76	24.07	1.55 ± 0.13	0.003 ± 0.001	0.002 ± 0.001	0.35 ± 0.03	12.9 ± 0.04
CS/Si 3	0.96	22.53	2.06 ± 0.15	1.13 ± 0.15	0.55 ± 0.09	0.38 ± 0.04	0.8 ± 0.02
CS/Si 4	1.12	21.27	0.36 ± 0.09	0.009 ± 0.001	0.02 ±0.001	0.37 ± 0.04	27.1 ± 1.44

## Data Availability

The data presented in this study are available on request from the corresponding author.

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
