# Peer review of "Mechanical, Structural, and Biological Properties of Chitosan/Hydroxyapatite/Silica Composites for Bone Tissue Engineering"

_molecules, 2021, doi:10.3390/molecules26071976_

Round 1

Reviewer 1 Report

The first thing that confused me was that the authors submitted this manuscript as a “review” while it is a typical experimental article.

Below are some questions/suggestions I have for the authors that should be addressed before accepting this manuscript as an article:

  1. The authors should give a more clear and concise introduction focusing on the main objective of this study. In the first two paragraphs, the authors spent quite a lot of words introducing that in recent decades, polymers have received great interest and have been widely explored. That is the general knowledge especially of bone tissue engineering, but it is not directly relevant to the present study, here the material is hybrid organic-inorganic CS/Si/HAp. So it should be given in a more concise way. The authors should give a more convincing introduction why the research is meaningful and important. For example, the authors should refer to previous studies and talk about what has been done and what have not been achieved.
  2. In the discussion I am missing reference and discussion of major work on mechanical, structural and biological properties of composites for bone tissue engineering, no reference to this substantial contribution to this area is given but furthermore, that research should be discussed in context to yours as it is highly relevant, for example Lett. 235, 207–211; Micron 2018, 119, 64–71; RSC Adv. 2016, 6, 66037–66047 but many others are available.
  3. The synthesis process of the hybrid CS/Si/HAp composites is not clear. For example, what was the pH? It was not given in the text. We are only told “0% acetic acid” was added. The authors used basic components such as hydroxyapatite and sol of silica (30% SiO2 and 0.36% Na2O) which react with acetic acid, so the pH had to change.
  4. Due to the fact that an acetic acid which was used in the reaction may react with basic hydroxyapatite, the authors should investigate whether HAp still exists in the mixture. At least XRD should be performed.
  5. In the experimental section the authors state that: :The test specimens were in the form of powder, without previous compression or using fluids such as KBr”. KBr is in a solid state, not a liquid.
  6. Density should be determined using a gas pycnometer.
  7. No bimineralization study was performed.

Author Response

Reviewer #1:

We are deeply grateful to the reviewer for taking the time to provide quite valuable comments and suggestions. We revised the manuscript in accordance with your advice.

1.The first thing that confused me was that the authors submitted this manuscript as a “review” while it is a typical experimental article.

Response: 

We have made a mistake. There should be „Article” instead of  „Review”.

Line 1, „Review” has been changed to „Article”.

2. The authors should give a more clear and concise introduction focusing on the main objective of this study. In the first two paragraphs, the authors spent quite a lot of words introducing that in recent decades, polymers have received great interest and have been widely explored. That is the general knowledge especially of bone tissue engineering, but it is not directly relevant to the present study, where the material is hybrid organic-inorganic CS/Si/HAp. So it should be given in a more concise way. The authors should give a more convincing introduction why the research is meaningful and important. For example, the authors should refer to previous studies and talk about what has been done and what have not been achieved.

Response: 

Thank you for your valuable comments. We hope that the slightly modified introduction is better.

To our knowledge, biocomposites based on chitosan, hydroxyapatite, inorganic silica, and calcium β–glycerophosphate or sodium β–glycerophosphate pentahydrate have not been studied yet by researchers. The present research is intended to fabricate porous hybrid bio-composite having optimal bioactive and mechanical strength for bone implants, based on chitosan (CS) and non-organic silica (SI), reinforced with calcium β–glycerophosphate (Ca-GP) or sodium β–glycerophosphate pentahydrate (Na-GP), and hydroxyapatite powder (HAp) which would be competitive in terms of price and quality. The composites obtained were characterized in terms of the chemical and mechanical properties through the Instron universal testing machine, scanning electron microscopy (SEM), and Fourier transform infrared (FTIR) spectroscopy. The biocompatibility of composites was also assessed. The influence of drying temperature on the mechanical properties of the developed composites was also studied.

3. In the discussion I am missing reference and discussion of major work on mechanical, structural and biological properties of composites for bone tissue engineering, no reference to this substantial contribution to this area is given but furthermore, that research should be discussed in context to yours as it is highly relevant, for example Lett. 235, 207–211; Micron 2018, 119, 64–71;RSC Adv. 2016, 6, 66037–66047 but many others are available.

Response: 

References and discussion have been added according to your suggestion.

  • Woźniak, M.J.; Chlanda, A.; Oberbek, P.; Heljak, M.; Czarnecka, K.; Janeta, M.; John, Ł. Binary bioactive glass composite scaffolds for bone tissue engineering—Structure and mechanical properties in micro and nano scale. A preliminary study. Micron 2019, 119, 64–71, doi:10.1016/j.micron.2018.12.006.
  • Baino, F.; Fiume, E.; Miola, M.; Leone, F.; Onida, B.; Verné, E. Fe-doped bioactive glass-derived scaffolds produced by sol-gel foaming. Mater. Lett. 2019, 235, 207–211, doi:10.1016/j.matlet.2018.10.042.
  • John; Janeta, M.; Rajczakowska, M.; Ejfler, J.; Łydzba, D.; Szafert, S. Synthesis and microstructural properties of the scaffold based on a 3-(trimethoxysilyl)propyl methacrylate-POSS hybrid towards potential tissue engineering applications. RSC Adv. 2016, 6, 66037–66047, doi:10.1039/c6ra10364b.

4. The synthesis process of the hybrid CS/Si/HAp composites is not clear. For example, what was the pH? It was not given in the text. We are only told “0% acetic acid” was added. The authors used basic components such as hydroxyapatite and sol of silica (30% SiO2 and 0.36% Na2O) which react with acetic acid, so the pH had to change.

Response: 

Thank you for your valuable comments. We have added the following sentences.

Line 109: The sentence has been added: „The source of inorganic silicon was sol of silica (Si, trade name Sizol 030) a commercial product of the “Rudniki” Chemical Works (Poland), essentially composed of 30% SiO2 and 0.36% Na2O in water, pH~9.”

In section 2.2.1. we have changed to the following:

„Briefly, 0,4 g of chitosan was dissolved in 10 g of 4.0% acetic acid. The solution obtained was stirred (under slow rotations) until complete dissolution for 1 h in a water bath at 55 °C. Next, 3.2 – 12.7 % wt. of hydroxyapatite was added gradually to the solution and stirred in an ultrasonic bath for 15 minutes to break down powder agglomerates. 5.5 g of CS/HAp mixture was added dropwise to 11 g of silica sol and vigorously stirred. During the synthesis, the pH values of the solutions oscillated between pH 4 and 8. Then, the paste obtained was poured into cylindrical PE molds with a diameter of 3 cm and height of 3 cm and aged for 24 h. Finally, CS/Si/HAp composites were dried in one day at 50 and 100 °C in the oven at atmospheric pressure. After drying, the cylindrical samples were weighed and measured. Following the fabrication process, the composites obtained were subjected to chemical, mechanical, and biological assessment.”

5. Due to the fact that an acetic acid that was used in the reaction may react with basic hydroxyapatite, the authors should investigate whether HAp still exists in the mixture. At least XRD should be performed.

Response: 

Unfortunately,  XRD research has not been done, but it is a valuable comment for the future. FTIR tests were performed.

6. In the experimental section the authors state that: :The test specimens were in the form of powder, without previous compression or using fluids such as KBr”. KBr is in a solid state, not a liquid.

Response: 

 The expression „using fluids such as KBr” has been removed.

7.Density should be determined using a gas pycnometer.

 Response: 

Lines 148-150: We have added the following sentences:

„Skeletal densities were determined using a helium pycnometer (AccuPyc 1330, Micromeritics Instrument Corporation, Norcross, Georgia, USA). Skeletal densities of samples were between 1.2 and 1.4 g/cm3.”

 8. No bimineralization study was performed.

 Response: 

In our preliminary studies, biomineralization study has not been performed to evaluate the bone regeneration capability from the relationship between the apatite formation in vitro and the ability to promote bone growth and regeneration in vivo.

We are planning to do it in the next step of research.

We hope that the revised manuscript better meets your expectations.

Reviewer 2 Report

This narrative review is under the scope of this journal; the topic is relevant for readers, and this research deals with potentially significant knowledge to the field.

  • However, there are some concerns about the present manuscript: 

Abstract

  • In the results, is important to show more information, for example, adds some values or p-values.

Introduction

Page 1- Line 39-41 need references for this sentence, please read this article, Palma et al. (DOI: 10.1016/j.joen.2013.10.023) investigated in an animal study the usage of chitosan scaffolds for dental pulp regeneration. However, it saw the CS was involved in the increase of bone tissue formation inside the root canal. Read this article https://doi.org/10.3390/polym12020436.

Results

  • How was the sample calculated? Did the authors perform a power analysis to evaluate if this sample size was appropriate?
  • Figure 1 and Figure 2a, please improve the image resolution.
  • Figure 6 – improve also the text caption and image resolution

Discussion

  • Please, clarified what was the limitation of this study?

And also, clarified the future perspectives also add in the discussion.

References

The titles of references have a different format, the title of the article is written in capital letters at the beginning of words, others only in lower case. Also, the standardized format of presentation in the journal's name. Because names have written in a different format, one is not abbreviated, others are not.

Author Response

Reviewer #2:

  1. This narrative review is under the scope of this journal; the topic is relevant for readers, and this research deals with potentially significant knowledge to the field.

Response: 

We have made a mistake. There should be „Article” instead of „Review”.

Line 1, „Review” has been changed to „Article”.

  1. Abstract

In the results, is important to show more information, for example adds some values or p-values.

Response: 

Thank you for your valuable comments. Important work achievement has been added to the abstract. The following sentences have been added: 

„The CS/Si/HAp/Ca-GP developed from all composites analyzed was characterized by the well-developed surface of pores of two sizes: large ones of one hundred μm and many smaller pores below 10 µm, whose behavior positively influenced cells proliferation and growth, as well as compressive strength in a range of 0.3 ÷ 10 MPa, Young's modulus 5.2 ÷ 100 MPa, and volumetric shrinkage below 60%. This proved to be a promising composite for applications in tissue engineering, e.g. filling small bone defects.”

  1. Introduction

Page 1- Line 39-41 need references for this sentence, please read this article, Palma et al. (DOI: 10.1016/j.joen.2013.10.023) investigated in an animal study the usage of chitosan scaffolds for dental pulp regeneration. However, it saw the CS was involved in the increase of bone tissue formation inside the root canal. Read this article https://doi.org/10.3390/polym12020436.

Response: 

According to your suggestion, we have added the following references to the paper:

Santos, J.M.; Palma, P.J.; Ramos, J.C.; Cabrita, A.S.; Friedman, S. Periapical inflammation subsequent to coronal inoculation of dog teeth root filled with resilon/epiphany in 1 or 2 treatment sessions with chlorhexidine medication. J. Endod. 2014, 40, 837–841, doi:https://doi.org/10.1016/j.joen.2013.10.023.

Huang, Y.M.; Lin, Y.C.; Chen, C.Y.; Hsieh, Y.Y.; Liaw, C.K.; Huang, S.W.; Tsuang, Y.H.; Chen, C.H.; Lin, F.H. Thermosensitive chitosan-gelatin-glycerol phosphate hydrogels as collagenase carrier for tendon-bone healing in a rabbit model. Polymers (Basel). 2020, 12, 1–15, doi:10.3390/polym12020436.

  1. Results

How was the sample calculated? Did the authors perform a power analysis to evaluate if this sample size was appropriate?

Response: 

The diameter, height, and weight of the wet composites were measured prior to placing the samples in the oven. Then, after drying the cylindrical samples were weighed and measured and finally were ground to obtain the desired dimension for morphological and compressive strength tests.

  1. Figure 1 and Figure 2a, please improve the image resolution.

Response: 

According to your suggestion, Figures 1 and 2a have been improved.

  1. Figure 6 – improve also the text caption and image resolution

Response:                   

According to your suggestion, Figure 6 has been improved.

  1. And also, clarified the future perspectives also add in the discussion.

Response: 

In the discussion section, sentences have been added:

In future research, biomineralization studies should be performed to evaluate the bone regeneration capability and drying strategies should be designed to achieve optimal physicochemical properties of the implants for bone tissue regeneration.

  1. References

The titles of references have a different format, the title of the article is written in capital letters at the beginning of words, others only in lower case. Also, the standardized format of presentation in the journal's name. Because names have written in a different format, one is not abbreviated, others are not.

Response: 

The titles of references and journal names have been reviewed and changed.

We hope that the revised manuscript better meets your expectations.

Round 2

Reviewer 1 Report

The authors improved their manuscript significantly, it is now suitable for publication and I recommend publishing this manuscript.

Author Response

Thank you.

Reviewer 2 Report

This review is under the scope of this journal; the topic is interesting for readers and this research deals with potentially significant knowledge to the field.

The authors improved the quality of the manuscript after the reviewer's indications. Congratulations!

However need correct a reference. Please correct the reference 21, because it is a different doi. Palma et al. (https://doi.org/10.1016/j.joen.2017.03.005) investigated in an animal study the usage of chitosan scaffolds for dental pulp regeneration. However, it saw the CS was involved in the increase of bone tissue formation inside the root canal. 

Palma, P.J.; Ramos, J.C.; Martins, J.B.; Diogenes, A.; Figueiredo, M.H.; Ferreira, P.; Viegas, C.; Santos, J.M. Histologic evaluation of regenerative endodontic procedures with the use of chitosan scaffolds in immature dog teeth with apical periodontitis. J. Endod. 2017, 43, 1279–1287. [Google Scholar] [CrossRef] [PubMed]

Author Response

The reference has been changed. 

Thank you.